# A randomised, controlled, observer-masked trial of corneal cross-linking for progressive keratoconus in children: the KERALINK protocol

Kashfia Chowdhury,[1] Caroline Dore,[1] Jennifer M Burr,[2] Catey Bunce,[3] Mathew Raynor,[4] Matthew Edwards,[4] Daniel F P Larkin[5]

[1]Comprehensive Clinical Trials Unit, University College London, London, UK
[2]School of Medicine, University of St Andrews, St. Andrew's, UK
[3]Primary Care and Public Health Sciences, Kings College London, London, UK
[4]Ophthalmology, Royal Hallamshire Hospital, Sheffield, UK
[5]NIHR Moorfields Clinical Research Facility, Moorfields Eye Hospital, London, UK

**Correspondence to**
Dr Daniel F P Larkin;
f.larkin@ucl.ac.uk

## ABSTRACT

**Introduction** The KERALINK trial tests the hypothesis that corneal cross-linking (CXL) treatment reduces the progression of keratoconus in comparison to standard care in patients under 17 years old. KERALINK is a randomised controlled, observer-masked, multicentre trial in progressive keratoconus comparing epithelium-off CXL with standard care, including spectacles or contact lenses as necessary for best-corrected acuity.

**Methods and analysis** A total of 30 participants will be randomised per group. Eligible participants aged 10–16 years with progressive keratoconus in one or both eyes will be recruited. Following randomisation, participants will be followed up 3-monthly for 18 months. The effect on progression will be determined by $K_2$ on corneal topography. The primary outcome measure is between-group difference in $K_2$ at 18 months adjusted for $K_2$ at baseline examination. Secondary outcomes are the effect of CXL on (1) keratoconus progression, (2) time to keratoconus progression, (3) visual acuity, (4) refraction, (5) apical corneal thickness and (6) adverse events. Patient-reported effects will be explored by questionnaires.

**Ethics and dissemination** Research Ethics Committee Approval was obtained on 30 June 2016 (ref: 14/LO/1937). Current protocol: V.5.0 (08/11/2017). Study findings will be published in peer-reviewed journals.

**Trial registration number** European Union clinial trials register (EudraCT) 2016-001460-11

## Strengths and limitations of this study

► This is the first randomised trial of corneal cross-linking (CXL) in keratoconus in children, in which group disease onset is at an early age, is perceived to be at high risk of progression to corneal transplantation and in which only observational studies have been published.

► A total of 60 patients aged 10–16 years with progressive keratoconus will be randomised to CXL or standard care including spectacles and contact lenses as required for best-corrected vision.

► The trial is designed to examine safety and efficacy of CXL in reducing progression, the primary outcome measure being between-group difference in $K_2$ at 18 months adjusted for $K_2$ at baseline examination and measured by masked optometrists.

► Secondary outcome measures at 18 months include keratoconus progression, visual acuity, refraction, adverse events and quality of life measurements.

► Follow-up to 18 months after randomisation is relatively short and any benefit found following CXL would require longer term analysis of efficacy.

## INTRODUCTION

Keratoconus is characterised by thinning and distortion of the cornea that results in visual loss from complex refractive error and corneal opacification. The prevalence in Europe has been reported as 1:1163[1] and 1:375.[2] The age at initial referral to hospital clinics is the second and third decade (mean age at diagnosis 28 years[2]), with progression until the early 30s in most affected eyes. In its early stages, keratoconus causes worsening of vision on account of increasing myopia and irregular astigmatism: spectacle correction provides good visual acuity in early disease only, until increasing irregular astigmatism requires correction with rigid contact lenses for best vision. Patients with more advanced keratoconus lose contact lens-corrected visual acuity on account of corneal opacification and corneal transplant surgery is eventually required in >20% of patients.[3] Keratoconus is often more advanced when first diagnosed in children than in adults, with faster subsequent disease progression.[4]

The most important parameters used in the assessment of keratoconus are the curvature of the cornea (presented as dioptre power (D)), apical corneal thickness in µm, refraction and best-corrected visual acuity. Earliest disease can be detected by corneal topography, which demonstrates thinning and irregularity of corneal curvature. Quantification of steepness of the corneal curvature in horizontal, vertical and multiple oblique

BMJ

meridians identifies the meridian of maximum corneal steepness ($K_2$) and the point of maximum steepness ($K_{max}$).

While the standard care described above involves treatment of the refractive consequences of keratoconus or replacement of the diseased cornea by a transplant, the concept of stabilising keratoconus and arresting its progression at a stage when there is still good unaided or spectacle-corrected vision is relatively recent. Corneal cross-linking (CXL) increases the stiffness of the cornea, which can arrest the progression of early keratoconus.[5] It is the only current intervention for this purpose. In the epithelium-off CXL procedure corneal epithelium is removed, riboflavin eye drops administered and the cornea exposed to ultraviolet (UV) light for 8 or more minutes. CXL has been reported to be effective in arresting keratoconus progression in the majority of treated adult eyes in a number of non-randomised studies (including Henriquez et al,[6] Hersh et al[7]) and randomised controlled trials (RCTs) (O'Brart et al,[8] Wittig-Silva et al.[9]). In the larger study by Wittig-Silva et al, a significant difference in progression of corneal power in the steepest axis (termed '$K_{max}$' by these authors but in later publications widely designated '$K_2$') between CXL and control eyes was reported: an improvement in CXL-treated eyes with flattening of $K_{max}$ by −1.03±0.19 D compared with an increase in $K_{max}$ for control eyes of +1.75 ± 0.38 D at 36 months. Adverse effects were not uncommon but mostly transient, including corneal oedema, superficial opacification and recurrent corneal erosions. Despite increasing information in relation to the efficacy of CXL a Cochrane Review conducted in 2015 concluded that evidence for the use of CXL in the management of keratoconus is limited due to the lack of properly conducted RCTs.[10]

In younger subjects, a number of observational studies of CXL in keratoconus patients <19 years have been published, each with limitations but each reporting effectiveness. Caporossi et al reported an uncontrolled study of 152 keratoconus patients ranging in age from 10 to 18 years, of whom follow-up post-CXL was available on only 61% of patients.[11] Inclusion criteria included several parameters which are well recognised to be characterised by intertest variability. In this treated patient group, a statistically significant reduction of $K_{max}$ by −0.4 D was found. Vinciguerra et al reported 40 CXL-treated eyes in patients with progressive keratoconus aged 9–18 (mean 14.2) years in a non-randomised prospective study.[12] Findings included improved visual acuity, reduced myopic spherical equivalent on refraction testing and flattening on keratometry readings compared with pre-CXL. Goodfrooij et al reported progression in 22% within 5 years of CXL.[13] Although the findings from these studies suggested a beneficial effect of CXL, more robust evidence is required to inform practice. Of note, no randomised trial has been undertaken in young patients. The KERALINK trial has been designed to investigate efficacy and safety of the established technique of CXL in progressive keratoconus in the paediatric age group, in which on account

of early disease onset there is such potential for keratoconus progression. This paper describes the design of the trial, which compares progression of keratoconus in a population of children and young patients randomised to CXL or standard care, and evaluates safety of the intervention in this patient group.

Evidence of effectiveness of CXL is of particular interest in young patients and has specifically been requested by the National Institute for Health and Care Excellence in the UK. KERALINK is a multicentre RCT in this patient group evaluating epithelium-off CXL, the technique of CXL which has been demonstrated to be effective in adults. If the trial demonstrates efficacy of CXL compared with standard care, and in particular if CXL is arrests keratoconus progression, this would have important implications for clinical management. Although we intend to follow-up the trial patients for several years after the proposed trial concludes in order to ascertain the duration of keratoconus stability, it is clear that arrested progression in a paediatric patient is likely (1) to obviate the need for contact lens correction and for later corneal transplant surgery and (2) to have correspondingly greater health and cost benefit than if the CXL were undertaken in adults. Trial findings will inform ophthalmologists, optometrists and inform future research and treatment policy.

## METHODS AND ANALYSIS
### Study design
KERALINK is a randomised controlled, observer-masked controlled trial in five centres in the UK. The study adheres to the tenets of the Declaration of Helsinki and is registered at www.controlled-trials.com and the European Union clinical trials registry. It was approved by the UK Health Research Authority, the Medicines and Healthcare Regulatory Agency and ethical approval was granted by the Brent Ethics Committee (reference 16/LO/0913). The trial is supervised by a trial management group (TMG), with independent oversight by a trial steering committee (TSC) and a data monitoring committee. Eligible patients are randomised in a 1:1 ratio to receive either CXL or standard care including spectacles or contact lenses as necessary (standard care of early keratoconus in the UK includes correction of refractive error and not CXL). Following randomisation, participants are followed for 18 months at 3-monthly intervals. Inclusion and exclusion criteria are shown in table 1. All follow-up measurements are performed by masked observers (optometrists) and the treating ophthalmologists are masked as to keratometry values on topography at follow-up. Randomisation commenced on 31 October 2016 and follow-up of the last recruited patient is estimated to complete in mid-2020.

### Definition of progression for eligibility
To differentiate true keratoconus progression from measurement artefact or minimal progression, an

**Table 1** KERALINK inclusion and exclusion criteria

| | |
|---|---|
| Inclusion criteria | Age at randomisation: 10–16 years |
| | Confirmed keratoconus diagnosis |
| | Progression on Pentacam topography in one or both eyes, steepest corneal meridian ($K_2$) or $K_{max}$ >1.5 D |
| Exclusion criteria | Apical scarring |
| | Cone apex thickness <400 µm |
| | $K_2$ >62.0 D or $K_{max}$ >70.0 D |
| | Rigid lens wear in both eyes and unable to abstain for 7 days pretopography examinations |
| | Down's syndrome |

increase on topography (Pentacam, Oculus GmbH, Wetzlar, Germany) in the steepest keratometry ($K_{max}$) or in the steepest corneal meridian ($K_2$) of at least 1.5 D was used as threshold for eligibility in one or both eyes. Based on this, eligibility was defined by an increase from baseline in $K_{max}$ or $K_2$ of >1.5 D between two topography examinations separated by 3 or more months. For each patient, the eye with the more advanced keratoconus at baseline will be categorised as the study eye for the primary analysis, unless that eye had undergone prior surgery such as corneal transplantation.

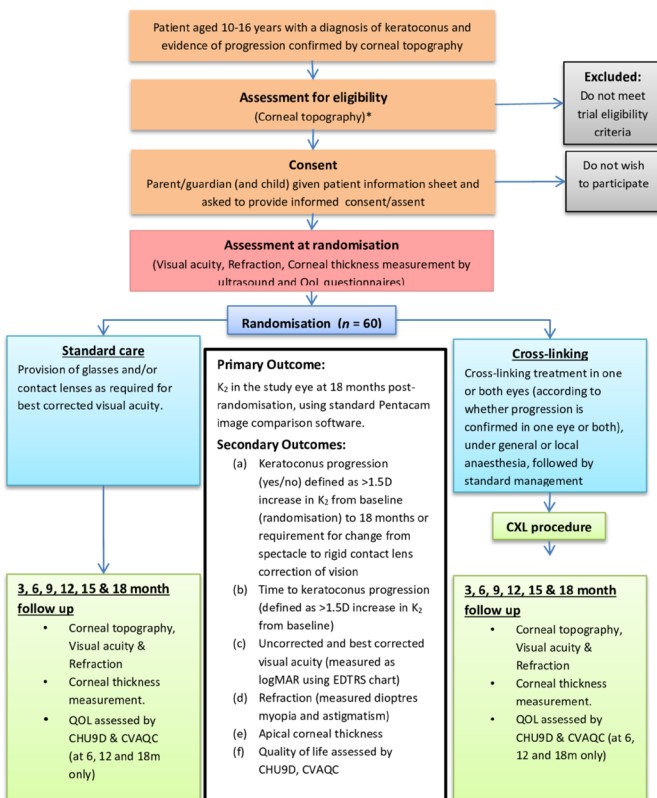

**Figure 1** KERALINK: efficacy and safety of cross-linking in children with keratoconus.

## Baseline assessment

At baseline all patients will be assessed as follows.

On these visits the following assessments will be performed.

1. Corneal topography for measurement of corneal power in the steepest meridian ($K_2$), used for assessment of the primary outcome. To improve repeatability, three measurements of each eye will be taken at baseline and follow-up examinations and the mean used in comparisons. Contact lenses will be removed at least 7 days prior to topography.
2. Visual acuity (unaided, spectacle-corrected and contact lens-corrected as applicable), logMAR measured using the Early Treatment Diabetic Retinopathy Study (ETDRS) chart at a starting distance of 4 m in both eyes.
3. Subjective refraction, both eyes
4. Apical corneal thickness measurement, both eyes, by ultrasound and Scheimpflug imaging at topography
5. Quality of life will be assessed by visual function (Cardiff Visual Ability Questionnaire for Children (CVAQC) and generic paediatric health outcome (Child Health Utility 9D (CHU9D) questionnaires. CVAQC is a 25-item vision-specific questionnaire designed for children.[14] CHU9D is a nine-question paediatric generic preference-based measure of health outcome which provides a descriptive health profile as well as a utility score and has been validated for self-completion in an adolescent population (11–17 years).[15]

## Randomisation and allocation of participants to treatment groups

Randomisation will be by a centralised computer-generated randomisation service (https://www.sealedenvelope.com). The system is customised to trial requirements, using minimisation with stratification by treatment centre and whether progression is confirmed in one eye or both eyes at randomisation. Following a dedicated consent/screening and randomisation visit for eligible patients and their parents, patients will be randomised to one of two trial arms (figure 1). Specific study information sheets will be provided to parents and patients prior to taking consent; a parent or guardian will be asked to provide consent in all cases and patients aged 15–16 years will be asked to provide assent if this is their choice.

## Intervention: CXL

Corneal cross-linking in one or both eyes (according to whether progression is confirmed in one eye or both), under general or local anaesthesia as applicable, followed by standard management. The surgical procedure will be as follows: insertion of lid speculum, removal of corneal epithelium with a spatula, administration of riboflavin drops (Vibex Rapid, Avedro, Waltham, Massachusetts, USA) every 2 min for 10 min, application of pulsed UV light using standardised parameters of 10 mW/cm$^2$ for a 5.4 J/cm$^2$ total energy dose administered over 8 min in a pulsed manner (Avedro KXL). At completion of

the procedure one drop of povidone iodine and a therapeutic contact lens will be applied to the treated eye. Management post-CXL is (1) proxymetacaine drops every 2 hours and naproxen 250 mg two times per day, both as required for analgesia, (2) moxifloxacin 0.5% drops every 6 hours for 1 week as infection prophylaxis, (3) dexamethasone 0.1% drops every 6 hours for 1 week, every 12 hours for 1 week, then fluorometholone 0.1% drops every 12 hours for 1 week. Patients randomised to CXL will attend for an extra examination at 1 week post-CXL for removal of the contact lens and confirmation of corneal re-epithelialisation.

### Comparator: standard care

The trial control arm is standard management alone, including refraction testing with provision of glasses and/or contact lens fitting for one or both eyes as required for best-corrected visual acuity.

### Defining keratoconus progression for secondary outcomes

To differentiate true keratoconus progression from measurement artefact, we will define progression as an increase in power in the steepest corneal meridian ($K_2$) of >1.5 D on corneal topography between two examinations or the requirement for change from spectacle to rigid contact lenses correction of vision, as the latter precludes reliable topography measurements.

### Outcome measures

The primary trial outcome measure will be between-group difference in $K_2$ at 18 months adjusted for $K_2$ at baseline examination.

Secondary outcomes will be the effect of CXL on
1. Keratoconus progression (yes/no) defined as >1.5 D increase from baseline in $K_2$, confirmed at subsequent visits *or* keratoconus progression requiring change from spectacle to rigid contact lens correction of vision, which precludes reliable topography measurements
2. Time to keratoconus progression.
3. Uncorrected and best-corrected visual acuity (logMAR) measured with an ETDRS chart at a starting distance of 4 m.
4. Refraction (measured dioptres spherical equivalent, myopia and astigmatism).
5. Apical corneal thickness.
6. Quality of life as assessed by paediatric health outcome and visual function questionnaires.

### Trial duration

All patients will be assessed at baseline, 3, 6, 9, 12, 15 and 18 months. Any patient found to have >1.5 D increase in $K_2$ will need to have this confirmed at a subsequent visit (ie, 3 months later). Participants who have unconfirmed progression at the 18-month follow-up visit will need this confirmed at a further visit at 21 months.

### Adverse events

Patients will be assessed for adverse events at the 1 week post-CXL follow-up and at all visits following randomisation.
1. Any reversible or short-term corneal abnormality, for example, prolonged eye pain, delayed corneal epithelialisation, transient corneal oedema.
2. Any visually significant corneal abnormality, for example, opacity resulting from sterile inflammatory infiltrates, corneal infection or stromal melting.
3. Any untoward medical occurrence in a study patient which does not necessarily have a causal relationship with the treatment under study, for example, abnormal laboratory findings, or disease symptoms and signs.

The Independent Data Monitoring Committee (IDMC) will monitor adverse events and serious adverse events during the trial to inform their recommendations to the TSC. Participants in the standard care arm with significant progression confirmed at two successive examinations will be considered for other keratoconus management options including cross-over to CXL

### Sample size calculation

The primary outcome is $K_2$ at 18 months, adjusted for $K_2$ at baseline, in the study eye recorded by an optometrist masked to the treatment group. A difference between the groups in the change in $K_2$ of >1.5 D from randomisation to 18 months is considered to be a clinically important difference (based on Wittig-Silva *et al*[9]). A $K_2$ increase >1.5 D would discriminate a true change in the steepest corneal meridian from measurement artefact and would be visually significant. A sample size of 46 patients would be required to detect this difference at the 5% significance level with 90% power, assuming a SD of 1.5 D. The total sample size has been increased to 60 patients (30 per group) to allow for up to 24% loss to follow-up. These estimates are based on 12-month and 24-month data reported by Wittig-Silva *et al* from which we estimated a pooled SD of the changes of 1.476 D. We expect that on average there will be 10% loss to follow-up in both groups. In the study by Wittig-Silva *et al*, 19% of patients withdrew, crossed over to CXL or had a transplant by 18 months. However, 18% of patients in the control group either received CXL or a transplant. If we specifically adjust the sample size to take account of 10% loss to follow-up and up to 20% of the control arm cross-over to CXL or transplant, then our planned total sample size of 60 patients would still provide at least 80% power to detect the clinically important difference. The trial protocol states that participants cannot cross over to CXL before 9 months.

### Patient and public involvement

Patients and parents were first involved in this research at a patient event hosted by Moorfields Eye Hospital. Topics on which opinions were collected included randomisation, cross-over and the duration of follow-up of trial patients. The research questions, design and trial outcome measures in the protocol were finalised following the

above meeting and additional input from the UK Keratoconus Self-Help and Support Association. This Association supported the trial by publicising the trial and by providing representatives on the TMG and the trial IDMC. The investigators will communicate a summary of the trial results to participants and their parents. The UK Keratoconus Self Help and Support Association will disseminate in their website and other communications the results to keratoconus patients. The burden of the intervention was discussed at our initial meeting with patients and parents and at the consent-taking stage in the trial.

## Statistical analysis plan

The primary analysis will be conducted following the intention-to-treat (ITT) principle where all randomised patients will be analysed in their allocated group whether or not they receive their allocated treatment. Patient characteristics at the time of randomisation will be summarised using mean and SD for continuous variables which are approximately normally distributed, median and IQR for variables which are not normally distributed, or by frequencies and percentages for categorical variables. All statistical tests will use a two-sided p value of 0.05 unless otherwise specified. All CIs presented will be 95% and two-sided. A detailed statistical analysis plan will be developed for approval by the TSC and review by the IDMC and finalised before the first statistical analysis of unmasked outcome measures. No formal interim analysis is planned, but reports concerning patient safety and key efficacy outcomes will be prepared for regular review by the IDMC who may request an interim analysis if a report raises concern. The IDMC is independent from the sponsor and funders. The membership, frequency of meetings, activity (including trial conduct and data review) and authority will be covered in the UCL CCTU IDMC terms of reference.

For each patient, the eye with the more advanced keratoconus at the time of randomisation will be defined as the study eye for the primary analysis, unless that eye has previously been treated by CXL or corneal transplantation. The analysis of the primary outcome will be performed using a linear mixed model fitted to all $K_2$ values recorded after randomisation. $K_2$ at randomisation, treatment group, follow-up time, the interaction between treatment and time, and the stratifying variables centre and whether each patient has one or both eyes eligible will be included as fixed effects. A random patient effect will be included to take account of clustering by patient. The regression coefficient for treatment group in this model estimates the difference between the mean changes in $K_2$ of each group.[16] Model assumptions will be assessed, and a logarithmic transformation may be used if this improves normality of the residuals. In the event of substantial (>10%) cross-over from the randomised arm to the other arm, we will perform two analyses of the primary outcome, the primary ITT analysis and a per-protocol analysis. The per-protocol analysis will exclude any information collected from a patient after cross-over.

Any cross-over or other treatment deviations will be summarised with reasons.

An ITT analysis will be performed for all secondary outcomes. Secondary continuous outcomes such as uncorrected and best-corrected visual acuity measured at randomisation and on more than one occasion during follow-up will be analysed using similar linear mixed models. Uncorrected and best-corrected visual acuity will be measured in logMAR using an ETDRS chart at a distance of 4 m. In patients for whom both eyes show progression at the time of randomisation, information from both eyes will be included in a secondary analysis including eye as a fixed effect and patient as a random effect.

Fisher's exact test will be used to compare the proportion of study eyes with keratoconus progression in each treatment group. Cox regression analysis will be used to estimate time to keratoconus progression in the study eye for each treatment group. The model will adjust for the stratifying variables, centre and whether each patient has one or both eyes eligible. Patients who do not progress during the course of the trial will be censored at their last follow-up visit.

We will also explore how visual disability and health in children and young patients with keratoconus relate to changes in $K_2$. The impact of missing data will be mitigated against by incorporating information from all observed time points using a mixed model approach.

Planned subgroup analyses will be conducted to investigate whether the effect of CXL differs between patients who had progression at randomisation in one or both eyes. This will be explored by adding an interaction between the number of eyes with progression at randomisation and CXL treatment to the primary efficacy outcome analysis mixed model.

# ETHICS AND DISSEMINATION
## Ethical and safety considerations

The trial was approved by the UK Health Research Authority and the Medicines and Healthcare Regulatory Agency. Trial investigators will ensure that the study (including any approved amendments) is conducted in accordance with the principles of the Declaration of Helsinki.

## Dissemination plan

The results of the trial will be reported in accordance with Consolidated Standards of Reporting Trials guidance and will be disseminated regardless of the direction of effect. Publications generated from the trial will be attributed to the TMG, which consists of all those who have wholeheartedly collaborated in the trial. The main report will be drafted by the TMG, and the final version will be reviewed by the TSC before submission for publication. Trial findings will be disseminated to the patients, UK Keratoconus Self-Help and Support Group and also doctors, optometrists, advisory bodies and healthcare commissioners. This will take the form of papers in peer-reviewed open-access medical journals and presentations at conferences.

**Acknowledgements** We thank Anne Klepacz and Mike Oliver of the UK Keratoconus Self Help and Support Association for their contributions to the KERALINK trial management group and independent data monitoring committee respectively.

**Contributors** DFPL was responsible for the trial concept. JMB, CB, MR and ME made substantial contributions to the design of the study and protocol. DFPL drafted the manuscript based on the KERALINK trial protocol. KC and CD drafted the statistical analysis methods, and all authors provided critical review and approved the final manuscript. Consent for publication is given by all authors.

**Funding** This work was supported by the Efficacy and Mechanism Evaluation Programme (reference 14/23/18), a MRC and NIHR partnership. The trial sponsor is University College London (contact Emilia.caverly@ucl.ac.uk). This research was otherwise supported in part by the NIHR Moorfields Biomedical Research Centre and the NIHR Moorfields Clinical Research Facility, London, United Kingdom.

**Disclaimer** This report presents independent research commissioned by the NIHR; the views and opinions expressed by authors in this publication are those of the authors and do not necessarily reflect those of the MRC, NIHR, the EME programme or the Department of Health.

**Competing interests** None declared.

**Patient consent for publication** Not required.

**Ethics approval** Ethical approval was granted by the Brent Ethics Committee (reference 16/LO/0913).

**Provenance and peer review** Not commissioned; externally peer reviewed.

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
