## [Reviewer comments · BMJ Open]

ARTICLE DETAILS

TITLE (PROVISIONAL)	A randomised, controlled, observer-masked trial of corneal cross-linking for progressive keratoconus in children: the KERALINK protocol
AUTHORS	Chowdhury, Kashfia; Dore, Caroline; Burr, Jennifer; Bunce, Catey; Raynor, Mathew; Edwards, Matthew; Larkin, Daniel

VERSION 1 – REVIEW

REVIEWER	Anders Ivarsen Department of ophthalmology, Aarhus University Hospital Denmark
REVIEW RETURNED	08-Mar-2019

GENERAL COMMENTS	This study protocol addresses a very important issue; whether collagen cross-linking reduces the risk of progression of keratoconus in comparison to standard care in children. Although a large amount of studies have found an effect of CXL in adults, properly conducted studies are few, and in general the effects of CXL are poorly investigated, especially so in children. Furthermore, progression of keratoconus is poorly understood and there is a tendency to believe that in children, progression of keratoconus occurs always and rapidly, although the scientific basis for this belief is lacking. The current study thus allows important questions to be examined, and being controlled, randomised, and blinded, it providing the best possible means for answering the raised questions. Study ethics are well accounted for, the design is sound, and the statistical approach is well described and appears to be comprehensive. On this basis I congratulate the investigators with a well design protocol, and look very much forward to seeing the results from the study.
---

REVIEWER	Elsie Chan Royal Victorian Eye and Ear Hospital, Australia
REVIEW RETURNED	07-Apr-2019

GENERAL COMMENTS	I commend the authors for establishing this RCT on CXL in the paediatric age group and I look forward to reading the outcomes of this study in due course. I only have minor comments to improve this manuscript. Note: page numbers refer to the numbering of the uploaded document (not the final PDF).
---

	 1. As the standard of care is considered to be CXL amongst many clinicians, the authors may consider changing the wording of 'standard care' to 'conventional care', or clarify that refractive correction (spectacles/ contact lenses) is still standard care in this age group in the UK 2. 'Refraction' – clarify if this is subjective refraction 3. Page 6, paragraph 1 – to clarify that the Kmax used in the Wittig-Silva study actually referred to K2 (as per this manuscript's definition) 4. Page 6, paragraph 1 – there are technically more than 2 RCT on CXL that have since been published (eg. Sharma et al, Int Ophthalmol 2015) . While it's not necessary to cite them all, it's worth rewording in a way to not imply there are only 2. The Sharma et al study has only 6 months follow-up, but it is interesting in that they actually performed a sham treatment with epithelial debridement. There is also the Hersh et al (Ophthalmology 2017) study. 5. Page 6, paragraph 2, line 1 – similarly, there are numerous studies on paediatric CXL although most are retrospective eg. Or et al, Cornea 2018 with 5 year follow-up. Can reword to not imply there are only 3 studies. 6. Page 7, paragraph 2 – 'In summary, evidence of effectiveness....'. I would remove the words 'in summary' as there is no prior mention of the NICE requirements in the introduction, and thus this paragraph does not represent a summary of previously mentioned statements. 7. Clarify how long contact lenses must be removed prior to eye examination, especially since best corrected visual acuity including contact lens-corrected visual acuity is documented (page 9, point 2) 8. How is apical corneal thickness measured? It is mentioned as ultrasound in the flow diagram but not in the text of the manuscript. 9. Page 9 – is slit lamp examination included in the assessments each visit? 10. Page 9 – 'intervention: CXL' – mention if a speculum is used (it is not by some operators). It should also be explained why 10mW/cm² for 8 minutes was chosen, as the majority of accelerated protocols use 9mW/cm² for 10 minutes (see Shajari et al. Acta Ophthalmol, 2018, a metaanalysis) . Is it because of manufacturer recommendations? Am I correct in assuming it's the Avedro KXL being used here as the Vibex rapid is used? The device details should be added, and whether it's continuous or pulsed ultraviolet UVA as I suspect the 8 minute treatment may be pulsed. Also, since povidone iodine is mentioned as being used post-CXL, is it used prior to treatment also? 11. Page 10 – 'outcome measures: change from spectacle to rigid contact lens correction of vision'. The timing and tolerance of this can be quite subjective. Was it based on a specified cut-off in terms of acuity? 12. As this is a study involving paediatric patients, the consent process should be described in the methods. And typographical queries:  1. Introduction, paragraph 2, should it be 'curvature of the cornea (presented as diopter power (D))'? 2. Table 1: cone apex thickness – missing the m for metres
--	--

REVIEWER	Amani E Badawi Mansoura Ophthalmic center-Egypt
-----------------	--

REVIEW RETURNED	27-Jul-2019
GENERAL COMMENTS	Good study, but long-term follow up is necessitated

VERSION 1 – AUTHOR RESPONSE

REVIEWER 1 COMMENTS

No amendment suggested.

REVIEWER 2 COMMENTS

1. Explanation that standard of care in UK does not at present mean CXL has been inserted in Methods, para 1.
2. 'Subjective' inserted in Methods, Baseline assessment.
3. Introduction para 3 amended to clarify use of term the Kmax by Wittig-Silva et al.
4. and 5. Introduction paras 3 and 4 reworded to incorporate the reviewers' helpful comments.
6. Introduction final para, first sentence amended as suggested.
7. Statement inserted in Basement assessment bullet point #1 indicating that lenses not worn for at least 7 days prior to topography.
8. Corneal thickness measured by ultrasound and topography (bullet point #4 amended).
9. In response to reviewer's question, slit-lamp examination of cornea was not done at each visit.
10. Speculum use has been added to 'Intervention: CXL' paragraph. UV at 10mW/cm² for 8 minutes was used according to manufacturer's guidance, as reviewer suspected. The CXL device details have been inserted as suggested. Povidone iodine drops were applied only at the completion of procedure (no amendment).
11. Change from spectacle to contact lens wear was not based on a specified cut-off in terms of acuity (no amendment).
12. As suggested more detail on taking of consent has been added to Methods section on Randomisation and allocation.
13. Both typos identified by the reviewer have been corrected.

We thank the reviewer for careful reading of the manuscript and helpful suggestions.

REVIEWER 3 COMMENTS

No amendment suggested.